

# Total water levels along the South Atlantic Bight during three along-shelf propagating tropical cyclones: relative contributions of storm surge and wave runup

Chu-En Hsu[1], Katherine Serafin[2], Xiao Yu[1], Christie Hegermiller[3], John C. Warner[4], Maitane Olabarrieta[1]

[1]Department of Civil and Coastal Engineering, University of Florida, Gainesville, FL 32611, USA
[2]Department of Geography, University of Florida, Gainesville, FL 32611, USA
[3]Sofar Ocean Technologies, San Francisco, CA 94105, USA
[4]U.S. Geological Survey, Woods Hole, MA 02543, USA

*Correspondence to*: Chu-En Hsu (chuen.hsu@ufl.edu)

**Abstract.** Total water levels (*TWL*s), including the contribution of wind waves, associated with tropical cyclones (TC) are
among the most damaging hazards faced by coastal communities. According to the report of the Intergovernmental Panel on
Climate Change (IPCC; Masson-Delmotte et al., 2021), TC–induced damages are expected to increase because of stronger TC
intensity, sea level rise, and increased populations along the coasts. TC intensity, translation speed, and distance to the coast
affect the magnitude and duration of increased *TWL*s and wind waves. Under climate change, the proportion of high–intensity
TCs are projected to increase globally (IPCC; Masson-Delmotte et al., 2021), whereas the variation pattern of TC translation
speed also depends on regions (Yamaguchi et al., 2020). There is an urgent need to improve our understanding of the linkages
among TC characteristics and *TWL* components. In the past years, hurricanes Matthew (2016), Dorian (2019), and Isaias (2020)
propagated over the South Atlantic Bight (SAB) with similar paths but resulted in different coastal impacts. We combined in
situ observations and numerical simulations with the Coupled Ocean–Atmosphere–Wave–Sediment Transport (COAWST)
modeling system to analyze the extreme *TWL*s under the three TCs. Model verification showed that the *TWL* components were
well reproduced by the present model setup. Our results showed that peak *TWL* depends mainly on the TC intensity, the
distance to the TC eye, and the TC heading direction. A decrease of TC translation speed primarily led to longer exceedance
duration of *TWL*, which may lead to more severe damage. Wave–dependent water level components (i.e., wave setup and wave
swash) were found to dominate the peak *TWL* within the near–TC wave field (60%). Our results also showed that in specific
conditions, the pre–storm wave runup associated with the TC–induced swell may lead to *TWL*s higher than at the peak of the
storm. This was the case along the SAB during Hurricane Isaias. Isaias's fast TC translation speed and the fact that its swell
was not blocked by any islands were the main factors contributing to these peak *TWL*s ahead of the storm peak.

## 1 Introduction

Total water levels (*TWL*s), defined as the combination of astronomic tides, mean sea level, storm surge, wave runup
(combination of wave setup and wave swash), associated with tropical cyclones (TC) are among the leading hazards faced by
coastal communities (e.g., Kalourazi et al., 2020; Sallenger, 2000). The Saffir–Simpson Hurricane Wind Scale (SSHWS) has





been used to estimate the potential impacts and damages caused by TCs based on the maximum sustained wind speed. However, the maximum wind speed, the TC translation speed (Liu et al., 2007; Xu et al., 2007), the size of the storm (Irish et al., 2008), and the storm track (Suh and Lee, 2018; Wang et al., 2020) affect wave heights, wave periods, and storm surge levels along the coast differently. Alipour et al. (2022) pointed out that using SSHWS as a proxy of the expected impacts alone

may lead to severe miscalculation, and they proposed a new scaling system associated with rainfall, storm surge, and wind speed. Irish and Resio (2010) proposed a hydrodynamics–based surge scale for hurricane surge hazard and an approach for predicting expected flood inundation and damages. Sallenger (2000) proposed a more complex approach in which the *TWL* relative to the dune crest ($D_{crest}$) and dune base ($D_{base}$) elevations was used to classify four expected morphological impact regimes: inundation, overwash, collision, and swash. In the swash regime, swash peak *TWL*s (including the 2% exceedance

swash amplitude) do not reach $D_{base}$. In the collision regime the peak *TWL*s exceeds $D_{base}$ but does not reach $D_{crest}$. In the overwash regime *TWL*s exceed $D_{crest}$ when the wave swash effects are accounted for. In the inundation regime *TWL*s exceeds $D_{crest}$ even without the effect of the wave swash. Coastal dunes experience the direct impacts of surf–zone processes in the inundation regime, when *TWL*s exceed $D_{crest}$. Thus, the inundation regime is expected to induce the most severe damages among the four impact regimes while the swash regime represents the least severe condition with less anticipated damage.

*TWL*s thus represent the combination of storm independent (the mean sea level and astronomic tides) and storm dependent (wave runup and storm surge) water level components, being a better indicator of the increased water levels than the storm surge alone (Stockdon et al., 2007). The wave runup is a wind wave dependent parameter composed by a wave-averaged sea level variation known as 'wave setup' and a wave–varying fluctuating component known as 'wave swash' (Stockdon et al., 2006). Previous efforts have shown the complexity and uncertainty of TC–induced surges and compound floods. However,

the response of *TWL*s to storm characteristics is more complicated than that corresponding to the storm surge, and the relative role of the wave runup and storm surge, and the dependency with storm characteristics are still not well understood.

There are primarily two approaches for computing *TWL*s during extreme storms: with numerical models (e.g., Hegermiller et al., 2019) and with observed water levels and waves (e.g., Serafin and Ruggiero, 2014). For example, $\eta_0$ (= astronomic tides + mean sea level + storm surge) observations were available at National Oceanic and Atmospheric Administration (NOAA)

tide gauges. Coupled ocean–wave modeling systems such as Coupled Ocean–Atmosphere–Wave–Sediment Transport model (COAWST; Warner et al., 2010) can also be applied to predict $\eta_0$ deterministically and probabilistically. However, the wave runup component needed to compute the *TWL*s is not captured by tide gauges, and ocean models usually do not have sufficient computational resolution in space and time to reproduce the wave setup accurately. Moreover, due to the use of phase-averaged models, coupled modeling systems such as ROMS–SWAN are not able to reproduce the wave swash component. While models

such as InWave (Infragravity Wave model, also installed within COAWST; Warner et al., 2018) can solve infragravity waves, models like XBeach (Roelvink et al., 2009) as a version to resolve the wave phase and simulate the wave swash. However, InWave and XBeach require higher resolution in space and time, which makes them not appropriate and not efficient for large spatial areas. To overcome the modeling challenge, the wave runup can be computed using empirical formulas and linearly added to $\eta_0$. For example, Serafin and Ruggiero (2014) applied the empirical formula proposed by Stockdon et al. (2006) to




compute the wave runup at NOAA tide gauges using the wave parameters at nearby National Data Buoy Center (NDBC) wave buoys along the U.S. West Coast. The empirical formula proposed by Stockdon et al. (2006) (Eq. 1 and Eq. 2) provides the 2% exceedance percentile of extreme wave runup ($R_2$):

$$R_2 = 1.1 \left( 0.35 \beta_f (H_0 L_0)^{\frac{1}{2}} + \frac{\left[ H_0 L_0 \left( 0.563 \beta_f^2 + 0.004 \right) \right]^{\frac{1}{2}}}{2} \right) , \quad 0.3 \leq \xi_0 < 4.0 , \tag{1}$$

$$R_2 = 0.043 (H_0 L_0)^{\frac{1}{2}} , \xi_0 < 0.3 , \tag{2}$$

in which foreshore beach slope ($\beta_f$) and deep–water wave parameters ($H_0$=deep water zero moment order wave height and $L_0$= deep water peak wavelength) and the Iribarren number ($\xi_0$) (Eq. 3) were required. $\xi_0$ was used to categorize wave breaker types (Battjes, 1974). In Eq. 2, the first part ($1.1 \cdot 0.35 \beta_f (H_0 L_0)^{\frac{1}{2}}$) represents wave setup, and the second part ($1.1 \cdot \frac{\left[ H_0 L_0 \left( 0.563 \beta_f^2 + 0.004 \right) \right]^{\frac{1}{2}}}{2}$) represents the combination of infragravity swash and incident swash. The foreshore beach slope is used in the calculation of wave runup and Iribarren number.

$$\xi_0 = \frac{\beta_f}{(H_0/L_0)^{1/2}} , \tag{3}$$

While beach slopes depend on local coastal morphology, wave heights and wavelengths also depend on storm characteristics. Stockdon et al. (2007) pointed out that the swash zone can be moved onshore along the beach profile due to the large waves and storm surges during extreme weathers. Consequently, the mean beach slope ($\beta_m$) was suggested and defined as the relevant slope in Eq. 1 and Eq. 3 during hurricanes. The deep–water wave parameters can be calculated by de-shoaling the waves from

a given point along the coast or shelf to deep water using the linear wave theory. The empirical formula developed by Stockdon et al. (2006) separated intermediate to wave–reflective beach scenarios ($0.3<\xi_0<4.0$, Eq. 1) from extremely dissipative conditions ($\xi_0<0.3$, Eq. 2). According to their dataset, $R_2$ under $\xi_0<0.3$ did not necessarily linearly depend on the beach slope and was generally dominated by infragravity waves. Thus, Stockdon et al. (2006) suggested to use a parameterization with a similar form for infragravity swash to model the $R_2$ under $\xi_0<0.3$ (Eq. 2). Although the field data employed by Stockdon et

al. (2006) did not specifically include highly energetic conditions during storms, Stockdon et al. (2014) showed that the model results had good agreement with the empirical formulation.

While $R_2$ in the Stockdon et al. (2006) formulation is represented by a linear increase with increasing $H_0$, Senechal et al. (2011) suggested an upper limit of $R_2$ at highly dissipative beaches under energetic conditions (e.g., storm). Senechal et al. (2011) proposed another empirical formula for $R_2$ based on $H_0$ alone (Eq. 4), to avoid the over–prediction under such scenarios.

$$R_2 = 2.14 \tanh(0.4 H_0) , \tag{4}$$



However, Senechal et al. (2011) stated that the saturation of $R_2$ required further studies and measurements under diverse beach scenarios before generalization. Despite the importance of the $R_2$ on *TWL* estimation, the sensitivity of $R_2$ to the choice of these formulas had not been thoroughly examined. While the observed data used in Stockdon et al. (2006) included part of the study area of the present work (i.e., North Carolina), most of the scenarios (>93%) at the peak TC–induced water levels along

the SAB during hurricanes Matthew, Dorian, and Isaias belonged to the intermediate condition, on which Stockdon et al. (2006) primarily focused ($0.3 < \xi_0 < 4.0$). Senechal et al. (2011) specifically considered the conditions during storm events. Thus, we employed these two empirical formulas for analysis and comparison.

The main goal of this study is to ascertain how TC characteristics affect the relative contribution of storm surge ($\eta_S$) and wave runup ($R_2$) to *TWL*s. TC characteristics are projected to change at global scale due to climate change and global warming

(Masson-Delmotte et al., 2021): the proportion of high–intensity TCs (i.e., SSHWS category 4 to 5) and the corresponding maximum sustained wind ($V_{max}$) are projected to increase. With that aim, we applied the COAWST modeling system to simulate the *TWL*s along the South Atlantic Bight (SAB; extending from North Carolina to Florida) during three historical TCs with similar tracks. In the recent past, three hurricanes (Matthew 2016, Dorian 2019, and Isaias 2020) propagated through the shelf of the SAB with similar tracks. The corresponding damages along the U.S. East Coast were $10.0 billion by Matthew

(Stewart, 2017), $1.6 billion by Dorian (Avila et al., 2020), and $4.8 billion by Isaias respectively (Latto et al., 2021). While the damage from Matthew was the highest from all storms, it was one order of magnitude higher than that of Dorian, even with similar wind speeds. Surprisingly, while Dorian had a stronger intensity than Isaias according to the SSHWS, Isaias caused more damage and the fastest $V_t$ across all three hurricanes within the SAB. How the differences in $V_{max}$, $V_t$, and distance to the coast influenced the *TWL* components during these three TCs was not well understood. With similar tracks over

the SAB, these three historical TCs provided the opportunity to determine the effects of each TC property on waves and *TWL* along the coast.

This paper is organized as follows: a brief review of the modeling system and setup applied in this work is presented following the introduction. Model verification based on the comparison with historical observations at six NOAA tide gauges can be found next. In the result section, *TWL* components along the SAB during Matthew 2016, Dorian 2019, and Isaias 2020 are

analyzed and compared. The applicability of the two empirical wave runup formulas and the effect of TC characteristics on wave runups are also discussed and presented.

**Table 1. Averaged values of TC parameters of the three historical hurricanes within the SAB (values obtained from the National Hurricane Center). $V_t$ is the translation speed of storms; $V_{max}$ is the maximum sustained wind; $P_{min}$ is the minimum atmosphere pressure; $R_{max}$ is the radius of maximum wind; the damages are estimated in billion USD.**

| Hurricane | $V_t$ (m s⁻¹) | $V_{max}$ (m s⁻¹) | $P_{min}$ (mb) | $R_{max}$ (km) | Damage (billion USD) |
|-----------|---------------|-------------------|----------------|----------------|----------------------|
| Matthew | 5.38 | 31.45 | 979.88 | 62.70 | 10.0 |
| Dorian | 3.61 | 34.65 | 975.43 | 64.12 | 1.6 |
| Isaias | 5.79 | 22.47 | 1001.20 | 57.68 | 4.8 |




## 2 Model description and setup

Following the modeling framework of Hegermiller et al. (2019), we configured COAWST as a coupled ocean–wave model and set it up to simulate the ocean and wave dynamics during hurricanes Matthew (2016), Dorian (2019) and Isaias (2020). Ocean dynamics were resolved with the Regional Ocean Modeling System (ROMS; Shchepetkin and McWilliams, 2005), while wind wave generation and propagation were simulated with Simulation WAves Nearshore (SWAN; Booij et al., 1999). The computational flowchart of the ocean circulation – wave coupling applied here is shown in the appendix (Fig. A1). The ocean and wave models used the same horizontal grids, with a 5 km resolution parent grid covering the entire U.S. East Coast and a 1 km resolution child grid covering the southern SAB.

### 2.1 Ocean model (ROMS)

COAWST used ROMS as its ocean circulation model. To solve the Reynolds Averaged Navier–Stokes equations (RANS) utilizing a three–dimensional terrain–following framework with a curvilinear coordinate transformation, a finite–volume approach was complemented on the numerical scheme of ROMS (Shchepetkin and McWilliams, 2005). Additional information on the forcing in the governing equations of ROMS is provided in Kumar et al. (2012), Warner et al. (2008b), and Warner et al. (2010).

### 2.2 Wave model (SWAN)

The third–generation spectral wave model SWAN (Booij et al., 1999) solved wave action evolution while considering refraction, shoaling, wave–current interactions, wind–wave generation, and varied wave energy dissipation (bottom friction, breaking, and white–capping). The semi–empirical formula derived from the JOint North Sea WAve Project (JONSWAP) was used to calculate bottom friction (Hasselmann et al., 1973). We calculated wind wave growth and white–capping using the formulas presented by Komen et al. (1984). We used discrete interaction approximation (DIA; Hasselmann et al., 1985) for the non–linear quadruplet wave–wave interaction.

### 2.3 Model coupling scheme

Using the Model Coupling Toolkit, water levels, current velocities, and wave fields are two–way coupled in COAWST (Warner et al, 2008a). A data exchange interval of 30 minutes between ROMS and SWAN, including water surface elevation, current velocities, wave heights, wavelengths, wave periods, and directions, was used by Hegermiller et al. (2019). This exchange interval was found to replicate nearshore hydrodynamics adequately and was applied in the present work. Specifics regarding the coupling method and an example case study are provided (Warner et al., 2008b; 2010). The wind shear stresses and sea surface roughness by Taylor and Yelland (2001) at the sea surface were computed and used to force the ocean model. The vortex–force formulation (Kumar et al., 2012; Uchiyama et al., 2010) was employed in the current study to account for



wave–current interaction. Furthermore, the wave and current boundary layer properties were estimated with the SSW_BBL
150  option, which used the Madsen model (1994).

## 2.4 Model setup

In the current work, winds, atmospheric pressure, relative humidity, and surface air temperature from the RAPid refresh (RAP)
reanalysis were employed to force ROMS (https://www.nco.ncep.noaa.gov/pmb/products/rap/). This dataset comprised
atmospheric pressure at mean sea level (MSL) and wind speeds 10 meters above MSL. Although RAP only covers a portion
155  of the computational domain, it has a spatial resolution of 13 km at hourly time intervals. The Global Forecast System (GFS;
50 km resolution with a 3–hour time interval; https://www.ncdc.noaa.gov/data-access/model-data/model-datasets/global-
forcast-system-gfs) provided wind and atmospheric pressure forces for offshore regions that RAP did not cover.

The U.S. East Coast domain has a horizontal grid resolution of 5 km with 896 ($\xi$–direction) ×336 ($\eta$–direction) grid cells. The
SAB domain had a horizontal grid resolution of 1 km with 272 ($\xi$–direction) ×376 ($\eta$–direction) grid cells. The numerical
160  grids of ROMS had 16 vertical layers. For the SAB grid and the U.S. East Coast grid, the baroclinic time steps in ROMS were
30 seconds and 15 seconds, respectively. To determine the initial conditions for the surface water levels, velocities, salinity,
and temperature, we used the re–analyzed data from the HYbrid Coordinate Ocean Model (HYCOM;
https://www.ncei.noaa.gov/thredds-coastal/catalog/hycom_region1/catalog.html). 13 tidal elements (M2, S2, N2, K1, K2, O1,
P1, Q1, MF, MM, M4, MS4, and MN4) from the TPXO Tide Model database at Oregon State University (Egbert and Erofeeva,
2002) were applied to the parent grid to simulate astronomic tides. The Flather boundary condition was applied at the
boundaries of the ROMS model (the northeast and southeast boundaries of the black–dashed box in panel A of Fig. A2 in the
appendix) for the momentum balance to radiate out deviations from exterior values at the speed of the external ocean waves.
A 2–day spin–up was done, followed by an 11–day simulation (i.e., 13 days in total). The initial conditions, such as currents,
water levels, temperature, and salinity, were examined to show that the 2–day spin–up is adequate for them to achieve the
equilibrium state in the model. It was determined that an 11–day simulation period, including at least 5 days prior to the storm's
peak, was sufficient to track the development and spread of swells near the SAB.

For the boundary conditions of the SWAN model for Hurricane Matthew, hourly bulk wave parameters (significant wave
height, mean wave direction, and peak wave period) from NOAA's WaveWatchIII re–analyzed global dataset
(https://polar.ncep.noaa.gov/waves/ensemble/download.shtml) were imposed at 47 boundary segments along the southeast and
northeast boundaries of the U.S. East Coast grid (the black–solid box in Fig. A2 in the appendix) assuming the JONSWAP
wave spectra NOAA's WaveWatchIII re–analyzed global dataset did not have available data during Dorian and Isaias. Thus,
we employed a larger grid to cover the North Atlantic Ocean and the Gulf of Mexico with our modeling system to generate
the wave boundary conditions for these two TCs for input as boundary conditions to the SWAN model. Wave spectrum was
solved with 60 and 25 directional and frequency bins. The parent and child grids were solved with 30 and 15 seconds as their
computational time steps, respectively. As for the atmospheric forcing, SWAN used the same GFS–RAP input as ROMS.





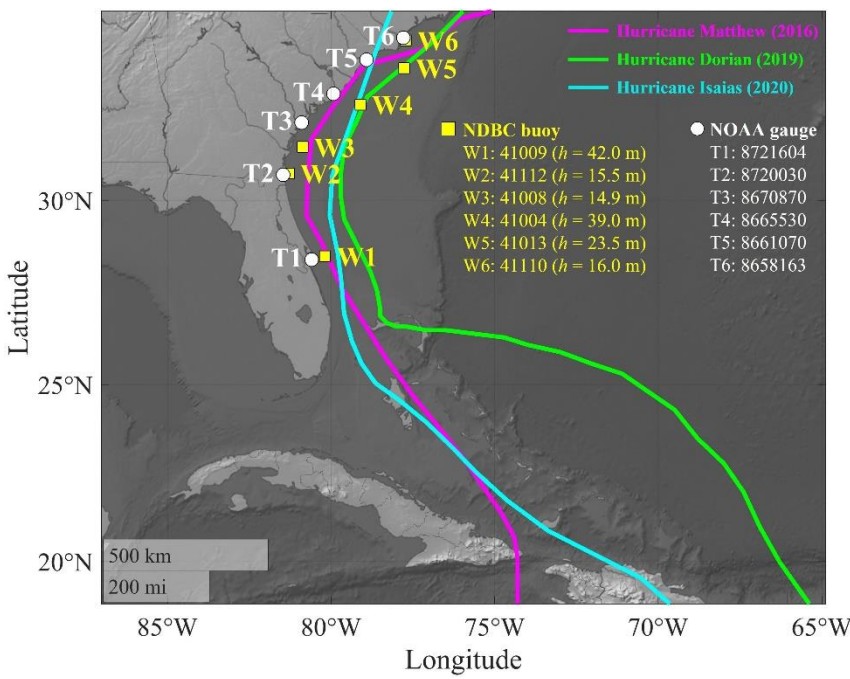

**Figure 1. NOAA tide gauges (white circles), NDBC wave buoys (yellow squares) selected for the model verification, and the paths of hurricanes Matthew 2016 (magenta), Dorian 2019 (green), and Isaias 2020 (cyan). *h* represents the water depth.**

**3 Model verification**

We used the method proposed by Serafin and Ruggiero (2014) to compute *TWL*s based on tide gauge and wave buoy observations. We used the observed data from six NOAA tide gauges and six NDBC buoys within the SAB to verify the model performance on $\eta_0$ and *TWL*s (Fig. 1). These NOAA tide gauges and NDBC buoys were selected based on the observation completeness during the three hurricanes and their locations (200 to 300 km away from each other). Tide gauges T1 to T4 are installed within estuaries, while T5 and T6 are installed at piers in local beaches. Thus, the measured $\eta_0$ at T1 and T4 may not

reflect the exact water levels at the beach. In this section, we used root–mean–square errors (*RMSE*) and the model skill (*skill*) proposed by Willmott (1981; Eq. 5) to quantify the model performances on $\eta_0$s and *TWL*s at six NOAA tide gauges.

$$skill = 1 - \frac{\Sigma_1^N |X_{model} - X_{obs}|^2}{\Sigma_1^N (|X_{model} - \overline{X_{obs}}| + |X_{obs} - \overline{X_{obs}}|)^2} \ , \tag{5}$$

where $N$ was the total number of data elements; $X_{model}$ was the model result; $X_{obs}$ was the observed data; the bar denotes time average.



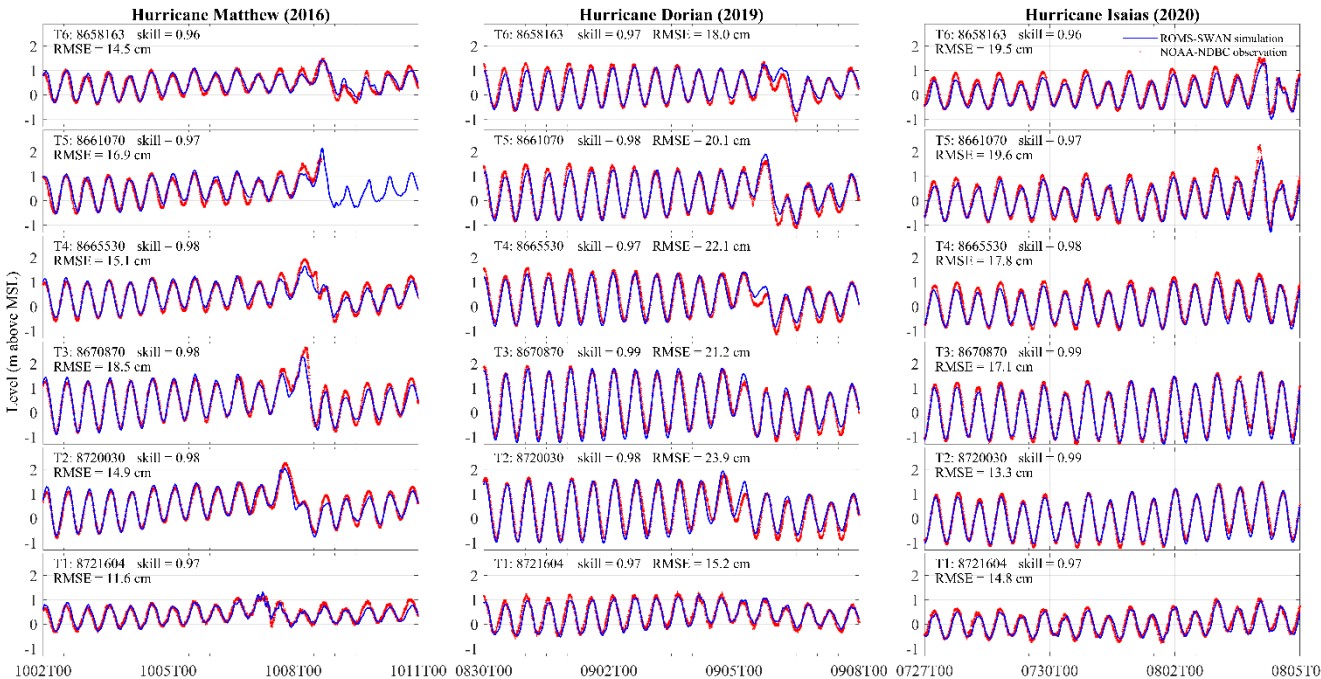

**Figure 2.** $\eta_0$ (= mean water level + astronomic tides + storm surge) time series at six NOAA tide gauges during the three historical hurricanes. *Skill*s were calculated with the formula of Willmott (1981), and *RMSE*s denoted the root–mean–square error between model results (blue curves) and observations (red points).

The predicted $\eta_0$ had *skill* = 0.95 on average (Fig. 2). The variation of $\eta_0$ strongly depends on the instantaneous tidal range. The tidal ranges during hurricanes Dorian and Isaias were 40 cm larger than that during Matthew on average among the six NOAA tide gauges.

Following the approach of Serafin and Ruggiero (2014), we used the measurements at NOAA tide gauges (T1–T6) and the nearby NDBC wave buoys (W1–W6) to compute *TWL*s including $R_2$ as the 'observational data' (red points in Fig. 3). The wave parameters (zero order moment wave height, $H_{m0}$, and peak wave period, $T_P$) at W1–W6 were used to compute the corresponding $R_2$ at T1–T6 using the formula of Stockdon et al. (2006). We used the linear wave dispersion relation to compute the representative deep–water peak wave parameters ($H_0$ and $L_0$). For model results, we used the predicted $H_{m0}$ and $T_P$ extracted at the COAWST computational grid with the shortest distance to the U.S. Geological Survey (USGS) data points along the SAB. The mean beach slopes measured by USGS before Hurricane Matthew along the SAB (Doran et al., 2015; Doran et al., 2017) were used to compute $R_2$ (Eq. 1). The *RMSE*s ($\leq$80.0 cm) and *skill*s ($\geq$0.80) of *TWL*s showed good agreement of the model results at the six NOAA tide gauges (averaged *skill*=0.93; Fig. 3). Particularly, the timing and the values of the peak *TWL*s during Matthew were well reproduced. Although USGS had some post–Matthew field surveys, these later measurements only covered a relatively small range or did not overlap with the SAB. To simplify the problem and to focus on the comparison of TC–induced water level components under the three historical TCs, the coastal morphology was assumed not to change between storms.



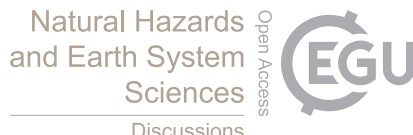

Overall, model results showed good agreement with NOAA observations: from the 36 analyzed time series of $\eta_0$ and *TWL*s at six stations during three storms, 32 of these time series had *skill*s higher than 0.90. The high *skill*s indicated that the numerically predicted total water levels and its components can be used to analyze the spatiotemporal variability of *TWL* components during the considered TCs. However, the lower *skill*s of *TWL*s compared to $\eta_0$ were related to the relatively poor performance of $T_P$ (Fig. A3 in the appendix).

Because of the potential influence of rainfall and river discharge on water level, especially at tide gauges T1 to T4 which are located within estuaries, we used the model results from the ROMS–SWAN model to analyze the storm–forced water level components at beaches along the SAB. This also allows for a higher spatial and temporal resolution of *TWL*s.

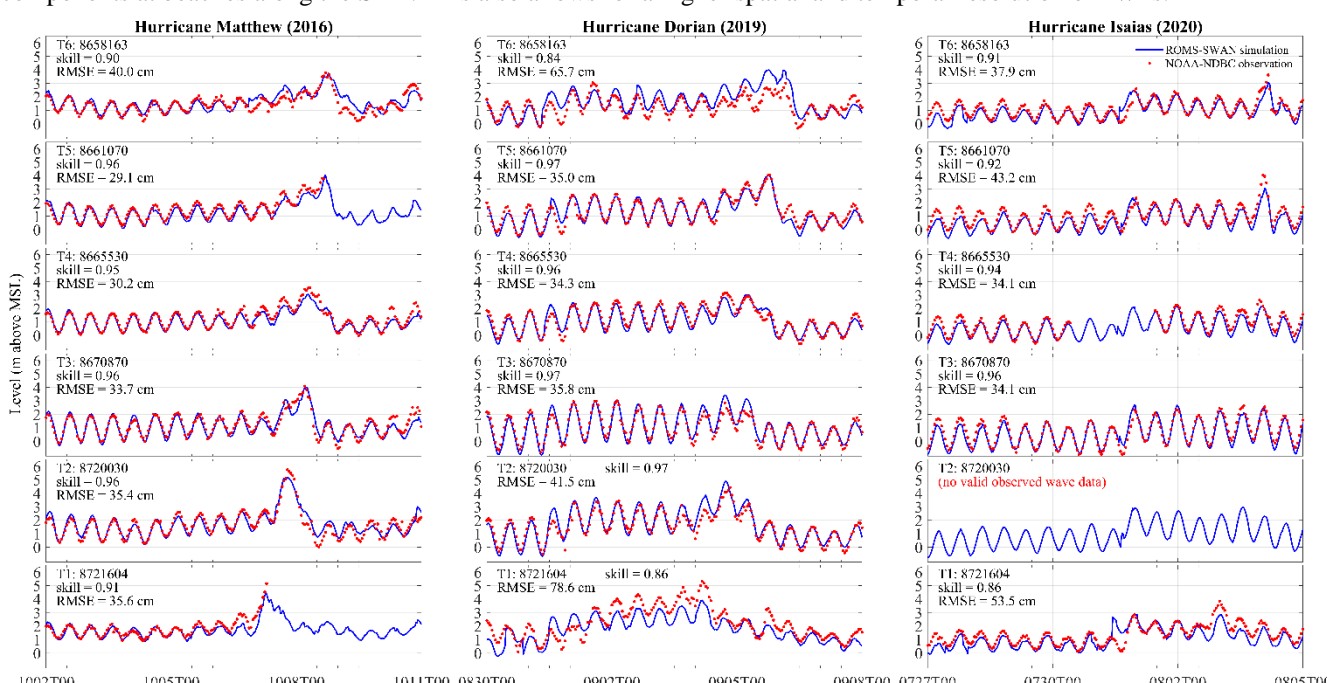

**Figure 3. *TWL*s time series at six NOAA tide gauges during the three historical hurricanes. *Skill*s were calculated with the formula**
**of Willmott (1981), and *RMSE*s denoted the root–mean–square errors between model results (blue curves) and observations (red points).**

## 4 Results

This section compares the time and spatial changes of the storm surge and wave runup along the SAB caused by hurricanes Matthew in 2016, Dorian in 2019, and Isaias in 2020. Additionally, the relationship between TC characteristics and the TC–
induced water level components is examined.


## 4.1 Storm forced water level components

*TWL* depends on the astronomic tides, which is at first order independent of TC characteristics, in such a way that the peak of the TC–induced water level can occur at any tidal level. Thus, to determine the influences of $V_{max}$, $V_t$ and TC path on the TC–induced water level components ($\eta_S$ and $R_2$), we utilized the Lanczos low–pass filter (Duchon, 1979) to separate $\eta_S$ from $\eta_0$.

After the separation of astronomic tides, model *skill*s decreased but were still higher than 0.85 at most considered tide gauges. As the storms approached the SAB, $\eta_S$ increased by more than 45 cm during Matthew and Dorian throughout all six NOAA tide gauges, while $\eta_S$ increased less than 30 cm at most of the NOAA tide gauges during Isaias (Fig. A4 in the appendix). We combined $\eta_S$ with the $R_2$ estimated by the Stockdon et al. (2006) formulation to obtain the TC–induced water level ($\eta_T$).

**Figure 4.** Top) Peak $\eta_T$ ($\eta_S + R_2$) along the SAB during the three hurricanes; $\mu$ represented the average and $\sigma$ was the standard deviation along the SAB. Bottom) re–analyzed GFS–RAP track data of the TC every six hours with the colormap presenting


instantaneous maximum sustained wind; the points with three colors (red, blue, and black) indicated the most severe levels achieved during the TC. We followed Sallenger (2000) and used local $D_{crest}$ and $D_{base}$ elevations to categorize the peak $\eta_T$. These categorizations were $D_{crest} \leq$ peak $\eta_T$; $D_{base} \leq$ peak $\eta_T < D_{crest}$; and peak $\eta_T < D_{base}$.

### 4.1 Peak values and durations of $\eta_T$ and $TWL$ over specified thresholds along the SAB

Matthew and Dorian had stronger $V_{max}$ on average (30.80 m s$^{-1}$ and 33.32 m s$^{-1}$ respectively) within the SAB compared to Isaias (22.50 m s$^{-1}$; Table 1), which led to lower surge levels during Isaias (15 cm to 90 cm lower than Matthew and Dorian). Meanwhile, Matthew's distance to the coastline (47.38 km) was the closest compared to Dorian (96.73 km) and Isaias (97.24 km) along the SAB on average. The sum of $\eta_S$ and $R_2$ gave $\eta_T$ along the SAB during the three historical hurricanes (top panel in Fig. 4). The peak $\eta_T$s along the SAB had similar distribution patterns during Matthew and Dorian, while the peak $\eta_T$ during Isaias was 60% to 65% smaller on average. The peak $\eta_T$s along Florida southeast coast were higher during Matthew compared to Dorian and Isaias, but the peak $\eta_T$s decreased significantly along Georgia and South Carolina as Matthew propagated northward and weakened. This led to a higher deviation of peak $\eta_T$s along the SAB during Matthew compared to Dorian and Isaias.

We used $D_{crest}$ and $D_{base}$ as thresholds to categorize the peak $\eta_T$ and $TWL$ along the SAB. This categorization referred to the morphological impact regimes of Sallenger (2000). The peak $\eta_T$ can occur coincidently with either the high tide or the low tide. Without the astronomic tides, the present work isolated and determined the contribution of TC–induced water level components and its dependency on TC characteristics. While Sallenger (2000) used the thresholds to categorize the morphological impacts caused by the $TWL$s, we utilized the thresholds to categorize the levels of TC–induced water levels ($\eta_T$) specifically. 23.0% and 19.9% of the coastal cites experienced peak $\eta_T \geq D_{crest}$ (red points in the bottom panels of Fig. 4) during Matthew and Dorian, respectively. These percentages were higher during Matthew and Dorian compared to those during Isaias (3.5%) (Table 2).

**Table 2. Percentage of coastal sites of each $\eta_T$ and $TWL$ categorization during the three historical TCs along the SAB.**

| | Categorizations of the peak $\eta_T$ | | | Categorizations of the peak $TWL$ | | |
|---|---|---|---|---|---|---|
| | $D_{crest}$ $\leq$ peak $\eta_T$ | $D_{base}$ $\leq$ peak $\eta_T$ $<D_{crest}$ | peak $\eta_T$ $< D_{base}$ | $D_{crest} \leq$ peak $TWL$ | $D_{base}$ $\leq$ peak $TWL$ $<D_{crest}$ | peak $TWL<$ $D_{base}$ |
| Matthew | 23.0% | 55.6% | 21.4% | 41.6% | 46.4% | 12.0% |
| Dorian | 19.9% | 54.8% | 25.3% | 42.0% | 49.2% | 8.8% |
| Isaias | 3.5% | 22.1% | 74.4% | 18.7% | 46.0% | 35.3% |

The proportion of coastal sites experiencing peak $TWL \geq D_{crest}$ was at least 1.8 times more than peak $\eta_T \geq D_{crest}$ (Table 2), which showed the importance of astronomic tides in coastal inundation levels. Matthew had a shorter distance to the coast



along the SAB compared to Dorian, while Dorian had stronger intensity north to Georgia (Fig. 4 and Table 1). Consequently, Matthew and Dorian induced comparable peak $\eta_T$s along the SAB.

In addition to the peak $\eta_T$, we used $D_{base}$ as the threshold to compute the maximum consecutive durations of $\eta_T \geq D_{base}$

($T_{ETA}$; top panels in Fig. 5) and $TWL \geq D_{base}$ ($T_{TWL}$; bottom panels in Fig. 5) along the SAB throughout each of the entire storm events. These were determined by calculating the maximum consecutive duration that $\eta_T$ or $TWL$ was higher than the thresholds without interruption. The $D_{base}$ was applied here because only <23% out of all coastal sites experienced peak $\eta_T \geq D_{crest}$ during the three TCs (Table 2). The averaged $T_{ETA}$ along the SAB during Dorian (55.1 hours) was longer than those during Matthew (32.5 hours) and Isaias (7.1 hours). Considering the contributions from astronomic tides, the averaged $T_{TWL}$

along the SAB were 27.5, 32.9, and 6.7 hours during Matthew, Dorian, and Isaias, respectively (Fig. 5). Note that the difference of averaged $T_{TWL}$ during Matthew and Dorian (i.e., 55.1-32.5=22.6 hours) was 76% smaller than the corresponding difference of averaged $T_{ETA}$ (i.e., 32.9-27.5=5.4 hours). This was mainly related to the smaller tidal range during Hurricane Matthew compared to Dorian. Although $TWL$ was larger than $\eta_T$ at high tides (crests of astronomic tidal signal), it was smaller than $\eta_T$ at low tides (troughs of astronomic tidal signal). This pointed out the importance of the instantaneous tidal range in the

inundation duration under extreme weather conditions.



**Figure 5. The maximum consecutive duration of $\eta_T$ ($T_{ETA}$; top panels) and $TWL$ ($T_{TWL}$; bottom panels) over $D_{base}$ at each USGS coastal site along the SAB throughout each of the three historical hurricanes.**

We calculated the maximum consecutive duration of $d/R_{max}$ (i.e., normalized distance) $\leq 8.0$ at each coastal site throughout

each of the three storm events, where $d$ was the distance between TC eye and each coastal site along the SAB and $R_{max}$ was

the instantaneous radius of maximum wind (Fig. 6). The threshold $d/R_{max} \leq 8.0$ followed the distance threshold of near–TC

wave field ($d/R_{max} \leq 8.0$ and $V_{max} \geq 33.0$ m s$^{-1}$) (Collins et al., 2018; Young, 2006). We did not consider the threshold of $V_{max}$

because $V_{max}$ did not reach 33.0 m s$^{-1}$ along the SAB during Isaias. We found this duration of $d/R_{max} \leq 8.0$ had a correlation

coefficient ($C_{CORR}$) = 0.47 with $T_{ETA}$ considering the coastal sites along the SAB during hurricanes Matthew, Dorian, and

Isaias. In particular, the durations of $d/R_{max} \leq 8.0$ during Matthew and Dorian (Fig. 6) showed similar patterns as $T_{ETA}$ (top

panel in Fig. 5). Meanwhile, the path of Hurricane Isaias had short distances to Florida southeast coast and resulted in the



duration of $d/R_{max}$ ≤8.0 longer than 48 hours. However, it did not lead to longer $T_{ETA}$, primarily because of the weaker $V_{max}$ of Isaias along the SAB.

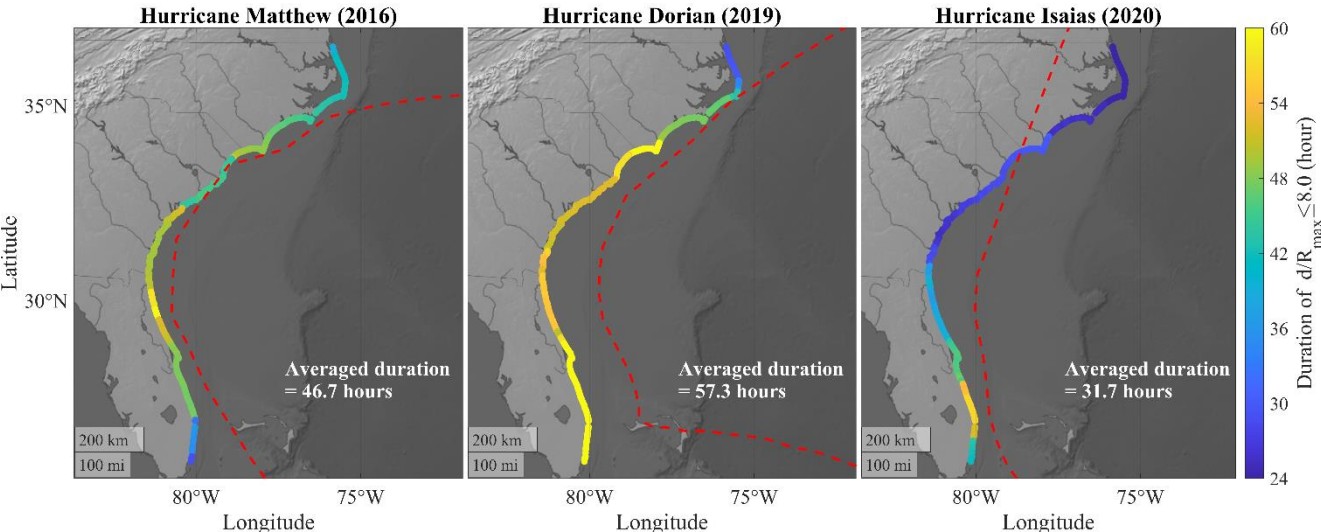

Figure 6. The durations of $d/R_{max}$ ≤8.0 along the SAB during three historical hurricanes. The red curves represented the historical track of each hurricane respectively.

## 4.2 The relative contribution of $\eta_S$ and $R_2$ to $\eta_T$

In addition to the peak $\eta_T$ along the SAB during the three historical hurricanes, we compared the proportions of $\eta_S$, wave setup, and wave swash at three specified locations: Edisto Island, South Carolina (32.51°N, 80.26°W); Sea Island, Georgia (31.20°N, 81.33°W); and the barrier island south of Matanzas Inlet, Florida (29.68°N, 81.22°W) (Fig. 7). Edisto Island, South Carolina ($D_{crest}$=2.10 m and $D_{base}$=1.26 m) and Sea Island, Georgia ($D_{crest}$=3.49 m and $D_{base}$=2.41 m) had relatively low dune elevations, in which dune overwash was more likely to occur during extreme weather events, according to the USGS (https://www.usgs.gov/news/national-news-release/fl-ga-sc-beaches-face-80-95-percent-chance-erosion-hurricane-matthew). The peak $\eta_T$ south of Matanzas Inlet, Florida during the storms were 1.41 m to 1.62 m (51% to 64%) greater than the other two locations in the near–TC wave field during Matthew and Dorian (timing shown by the vertical black dash lines in Fig. 7). One of the factors causing higher estimated $R_2$ was the larger mean beach slope south to Matanzas Inlet (0.151) compared to Sea Island (0.038) and Edisto Island (0.048) (Fig. A5 in the appendix). $R_2$ consisted of wave setup and wave swash. The percentage of wave swash in the peak $\eta_T$ outnumbered that of wave setup by 25% to 34% at Sea Island and Edisto Island during all three TCs, while the swash only outnumbered wave setup by less than 9% at the barrier island south of Matanzas Inlet (Fig. 8). Meanwhile, we found that $\eta_S$ contributed to less than 40% in the peak $\eta_T$ at the three locations as these three historical TCs approached. Surge levels at the peak $\eta_T$ generally decreased from south to north during the three hurricanes, whereas wave setup and wave swash did not experience such a pattern.





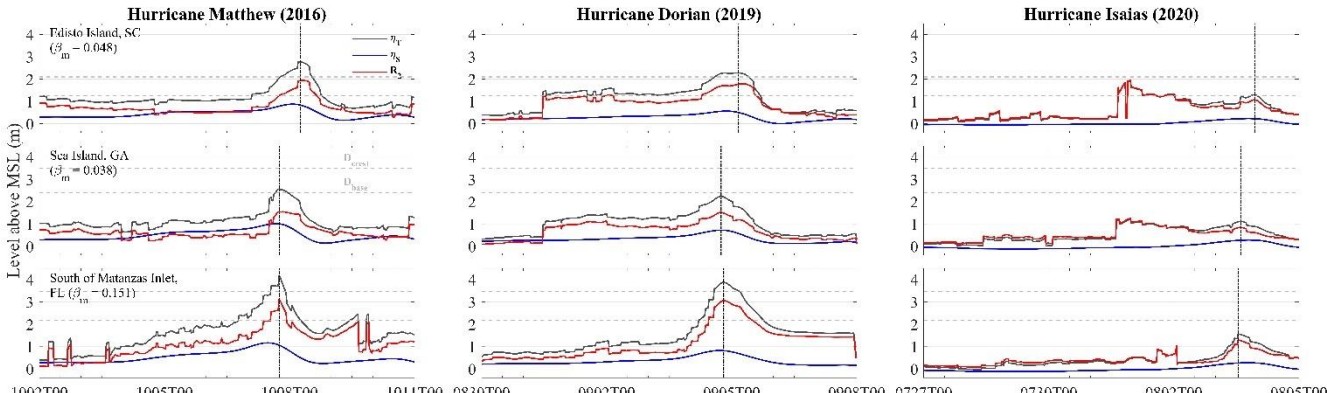

**Figure 7.** The time series of $\eta_T$ (black curves), $\eta_S$ (blue curves), and $R_2$ (red curves) at three selected locations during the three historical hurricanes. The horizontal gray dash lines were the local $D_{crest}$ and $D_{base}$ measured by USGS before Matthew (2016), and the vertical black dash lines were the peak $\eta_T$ in the near–TC wave field.

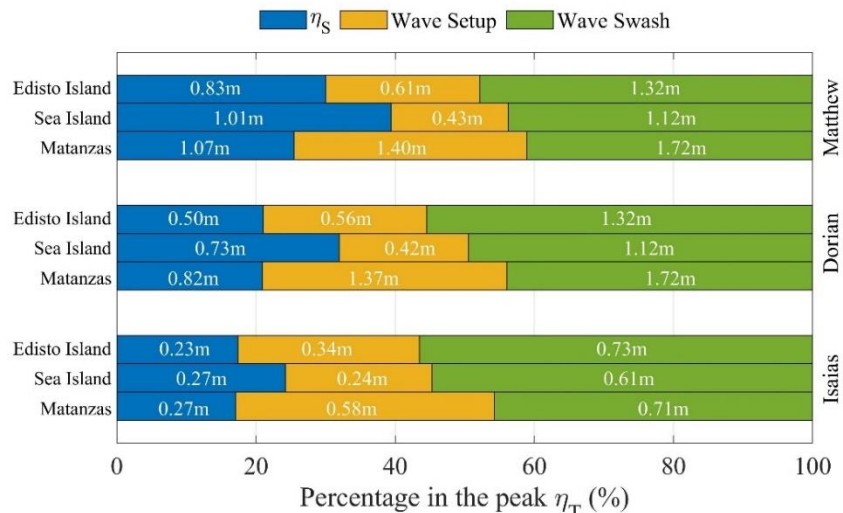

**Figure 8.** The contributions of $\eta_S$, wave setup, and wave swash in the peak $\eta_T$ during the three historical hurricanes at the three coastal sites: South of Matanzas Inlet, Florida; Sea Island, Georgia; and Edisto Island, South Carolina with the corresponding contribution of each component (%) in the peak $\eta_T$ and the level listed by the number written in white font.

Within the near–TC wave field, waves in most frequency bands kept receiving energy from the local wind, and $\eta_T$ was directly impacted by the instantaneous TC characteristics. The peak $\eta_T$ occurred within the near–TC wave field between 15:00 UTC 07 October 2016 and 07:00 UTC 08 October 2016 during Matthew, while it took place between 14:00 UTC 04 September 2019 and 04:00 UTC 05 September 2019 during Dorian (Fig. 7). The peak $\eta_T$ at the three locations occurred in the near–TC wave field during Matthew and Dorian, without a second comparable peak $\eta_T$ throughout the time series. The scenarios during Isaias were unique and different from Matthew and Dorian. First, the instantaneous $V_{max}$ did not reach 33 m s$^{-1}$ during Isaias when the normalized distance ≤8.0. Second, $\eta_T$ generally experienced an abrupt increase before Isaias's approach along the





coastal sites along the SAB, and this earlier increase of $\eta_T$ at Edisto Island, South Carolina even exceeded the peak value within the near–TC wave field (Fig. 7). This increased $\eta_T$ before the peak of the storm occurred between 21:00 and 23:00 UTC

31 July 2020 at Sea Island and Edisto Island, when Isaias was still far away from these three selected locations (1300 km).

$\eta_S$ and $R_2$ at the coast depended on the instantaneous TC characteristics within the near–TC wave field. The differences between the peak $\eta_T$ within the near–TC wave field during hurricanes Matthew and Dorian were less than 1.0 m at the three selected locations. The peak $\eta_T$ at the same locations during Hurricane Isaias within the near–TC wave field (03 August 2022 UTC) was 1.0 m to 2.7 m less than that of Matthew and Dorian. The $\eta_S$ during Isaias was 50% to 80% lower than that of

Matthew and Dorian within the near–TC wave field at the three locations (referring to the numbers listed in Fig. 8). Meanwhile, the peak $R_2$ (setup+swash) during Isaias was 40% to 60% smaller compared to that of Matthew and Dorian in the near–TC wave field. This was related to Isaias's smaller $V_{max}$ within the SAB (28% to 36% smaller than Matthew and Dorian; Table 1).

**Table 3. Maximum continual durations of $\eta_T$ and $TWL$ over the local $D_{base}$ at the three barrier islands during the three historical TCs in hours. $T_{ETA}$ was the maximum continual duration of the scenario $\eta_T \geq D_{base}$; $T_{TWL}$ was the maximum continual duration of the scenario $TWL \geq D_{base}$.**

| | $T_{ETA}$ (hour) | | | $T_{TWL}$ (hour) | | |
|---|---|---|---|---|---|---|
| | | | South of Matanzas | | | South of Matanzas |
| | Edisto Island | Sea Island | Inlet | Edisto Island | Sea Island | Inlet |
| Matthew | 40.5 | 7.5 | 35.5 | 31.0 | 10.5 | 23.0 |
| Dorian | 125.0 | 0.0 | 37.0 | 24.5 | 6.0 | 24.0 |
| Isaias | 16.5 | 0.0 | 0.0 | 6.0 | 0.5 | 1.5 |

The duration of the same $\eta_T$ category ($\eta_T \geq D_{crest}$; $D_{base} \leq \eta_T < D_{crest}$; and $\eta_T < D_{base}$) varied with TC characteristics. Similarly, $T_{ETA}$ at Edisto Island lasted up to more than five days during Dorian, which was much longer compared to the $T_{ETA}$

during Matthew and Isaias (40.5 hours and 16.5 hours, respectively; Table 3). $T_{ETA}$ at the barrier island south of Matanzas Inlet during Dorian (37.0 hours) was longer compared to Matthew (35.5 hours), but the difference was smaller than that at Edisto Island. While $T_{ETA}$ depended on TC characteristics alone, $T_{TWL}$ also depended on the instantaneous local tidal range. The *TWLs* at tidal troughs were lower with a larger tidal range. This led to a shorter $T_{TWL}$ as *TWLs* dropped lower than $D_{base}$ at tidal troughs. Although Dorian had a stronger $V_{max}$ and slower $V_t$ along the SAB on average, Matthew and Isaias had shorter

distances to the coast. Moreover, the tidal range during Matthew was 40 cm smaller compared to Dorian and Isaias on average at the six NOAA tide gauges shown in Fig. 1. With similar $V_{max}$, shorter distance to the coast and a smaller tidal range, Matthew had longer $T_{TWL}$ at Edisto Island and Sea Island compared to Dorian (Table 3).





## 5 Discussion

Suh and Lee (2018) utilized two historical TCs to analyze and compare the propagation processes of forerunner surges and
primary surges in Yellow Sea, and these processes were linked to the heading direction, path, and translation speed of the
storm. Similarly, we observed distinct patterns of storm–dependent water–level components variations during three different
storm events and at various locations along the SAB. These patterns are discussed in the following sections. Additionally, an
assumption of invariant coastal morphology between storm events was adopted due to limited data and to focus on analyzing
the TC characteristics effects on water level components. How this simplification affected the estimated wave runup is
discussed in the following section as well.

### 5.1 Storm forced surface water level variation

During Matthew and Dorian, the peak $\eta_T$s occurred when the coastal sites started to be covered by the near–TC wave field,
which was induced by the wind waves and $\eta_S$ associated with higher TC intensities (i.e., larger pressure deficits and higher
wind speeds). However, the $\eta_T$s at the three coastal sites had another local maximum at 15:30 UTC 31 July 2020 during Isaias,
when the storm was still located around 21.5°N, i.e., south of the SAB (Fig. 9b and Fig. 9d). This was primarily the result of
two factors. First, before entering the SAB (i.e., south of 26.0°N and east of 79.0°W), the translation speeds of Matthew
(maximum of 7.52 m s⁻¹ and average of 4.17 m s⁻¹) and Dorian (maximum of 7.35 m s⁻¹ and average of 5.35 m s⁻¹) were slower
compared to Isaias (maximum of 9.81 m s⁻¹ and average of 6.74 m s⁻¹). Xu et al. (2007) found that the swell energy and
wavelength increased when $V_t$ was comparable to the group wave celerity and under 13 m s⁻¹. This allowed wind waves to
experience an extended wind fetch and resulted in the growth of wavelength and wave height. According to Eq. 1 (Stockdon
et al., 2006), $R_2$ increases with the deep–water peak wavelength and the deep–water significant wave height. Second, before
arriving at the Island of Hispaniola (19.0°N), the swell generated by Matthew on its right–hand side was blocked by the Island
of Hispaniola and was unable to propagate toward the SAB on its path. By contrast, the swell generated by Isaias on its front–
right quadrant was not blocked by any island due to its path (see Fig. 1). Thus, the condition during Isaias was better for swell's
wavelength to be lengthened and to propagate ahead of the storm.



**Figure 9. The propagation of the swells generated by Hurricane Isaias on its right-hand side. The green stars indicated the three selected barrier islands, i.e., Edisto Island, Sea Island, and the barrier island south of Matanzas Inlet (from North to South). The red triangles represented the eye of Isaias with the red circle denoting the instantaneous radius of maximum wind and the red arrow denoting the heading direction. The black arrows were the mean wave directions from COAWST simulation. The colormaps in panels (a) and (b) showed the distribution of $H_{m0}$ at 03:30 UTC 31 July 2020 and 15:30 UTC 31 July 2020, respectively. The colormaps in panels (c) and (d) showed the distribution of $T_P$ at 03:30 UTC 31 July 2020 and 15:30 UTC 31 July 2020, respectively.**

According to the model results and linear wave dispersion relation, the peak wave period was 19.1 s and the deep–water phase celerity was 25 m s$^{-1}$ to 30 m s$^{-1}$ at Sea Island, Georgia at 15:30 UTC 31 July 2020 during Isaias (when the abrupt elevated $R_2$ occurred). This swell with a relatively long wave period generated by Isaias on its right–hand side arrived at the SAB coast





much earlier (one to two days) than the storm, which led to an abrupt increase in $R_2$. Around 16:00 UTC 01 August 2020, the instantaneous $V_t$ of Isaias decreased from 7.0 m s$^{-1}$ to <4.5 m s$^{-1}$. Additionally, waves with different periods travel with different phase celerities according to linear wave dispersion relation. This is also consistent with the distribution pattern of peak wave periods shown in Fig. 9c and Fig. 9d (i.e., waves with higher $R_2$ moved forward faster and approached the SAB

earlier). Consequently, the wavelength of the swell arrived later at the SAB decreased, which led to a decrease of $R_2$.

## 5.2 Coastal impact regimes of Sallenger (2000) and the temporal variation of $\beta_m$

The time–invariant dune elevations measured by USGS before Matthew did not reflect the realistic conditions during Dorian and Isaias, since the beach morphology (e.g., dune heights and beach slopes) changed in time. However, the time-invariant dune elevations allowed the present work to focus on determining the relative contributions of TC–induced water level

components ($\eta_S$ and $R_2$) during various TCs. The coastal impact regimes (Sallenger, 2000) were determined with the relative *TWL*s dependent on storm forced parameters ($\eta_S$ and $R_2$), astronomic tides, and coastal morphology (dune elevations and beach slopes). However, the actual coastal impact regimes during specific events required an update of the beach slopes and dune elevations. The problem was that this information was not always available at the spatial scale of this study.

To determine the sensitivity of the wave runup to the beach slopes, we used the post-Matthew $\beta_m$ from Georgia to North

Carolina measured by USGS (Doran et al., 2017) to compare *TWL*s and $T_{TWL}$ associated with the model results of Hurricane Dorian with the pre–Matthew surveyed $\beta_m$ (Fig. 10). We used the post–Matthew beach morphological information to determine the difference in the estimated storm–induced water levels. The pre–Dorian beach morphology can be different from the post–Matthew data we applied. However, the goal is to determine the effects of changing beach morphology on the storm-dependent water level components. The post–Matthew dataset showed that $\beta_m$ experienced an averaged decrease of −0.026

during Hurricane Matthew. According to Eq. (1) and Eq. (2), *TWL*s and $T_{TWL}$ during Hurricane Dorian may experience decreases considering the change of $\beta_m$. Results showed that 50% of the coastal sites between Georgia and North Carolina experienced an absolute difference of simulated peak *TWL* <0.5 m. The averaged decrease of peak *TWL* was 0.56 m with a standard deviation 0.87 m. Meanwhile, an averaged decrease of $T_{TWL}$ of 11.23 hours was observed (Fig. 10).

Based on Sallenger's (2000) categorization, our model results showed that the *TWL* ($\eta_T$ + astronomic tides) scenarios belonged

to the overwash and collision regimes during Matthew and Dorian, respectively, under the assumption of constant dune elevations and $\beta_m$. However, dune heights and $\beta_m$ are expected to decrease after storm events, which is consistent with the post–Matthew conditions from the observed data. While a decreased dune elevation led to more severe impact regimes, a milder $\beta_m$ resulted in a lower wave runup.





**Figure 10. Histograms (top panels) and spatial distributions (bottom panels) of the differences of $\beta_m$, peak *TWL*, and $T_{TWL}$ under pre– and post–Matthew conditions from Georgia to North Carolina coasts: A) $\beta_m$; B) peak *TWL*; C) $T_{TWL}$.**

### 5.3 Arrival timing of the peak storm–dependent components

Beside beach morphology and TC–dependent parameters ($\eta_T = \eta_S + R_2$), TC–independent parameters like astronomic tides also influenced the coastal impact regimes. Take the barrier island south to Matanzas Inlet, Florida for example, we found that the peak *TWL* exceeded $D_{crest}$, while the $\eta_T$ did not exceed $D_{crest}$. This was related to the coincidence of the timing of the high tide and the peak $\eta_T$. In the case that the peak $\eta_T$ occurred at low tide, the peak *TWL* would be lower than the peak $\eta_T$. With the time–invariant dune elevation, both the peak *TWL* and the peak $\eta_T$ did not reach $D_{crest}$ at this location during Dorian. However, both peak *TWL* and peak $\eta_T$ would exceed $D_{crest}$ in case that the dune elevation became lower.



Natural Hazards
and Earth System


### 5.4 Effects of TC properties on the durations of $\eta_T$ components

Hurricane Dorian travelled with a faster $V_t$ (3.61 m s$^{-1}$) on average, which was not only slower than the other two historical storms but also slower than the global average of TCs in all categories (4.20 m s$^{-1}$ to 6.00 m s$^{-1}$). While the instantaneous $V_{max}$ had significant impacts on the $\eta_T$, the slow movement of the TC resulted in a longer duration of $T_{ETA}$ at which a specific location was under its impact. The peak $\eta_T$ did not reach $D_{base}$ ($T_{ETA}$=0.0) at Sea Island during Dorian while the peak $\eta_T$ exceeded $D_{base}$ for 6.0 hours during Matthew. $V_t$ was the primary factor determining $T_{ETA}$ at the other two coastal sites.

However, as $V_{max}$ increased and/or the distance to the TC eye decreased, $T_{ETA}$ may also increase. The variations of $T_{ETA}$ at the three coastal sites during the three historical TCs implied that the peak water level alone may not be sufficient to predict the coastal impacts in practical scenarios. For instance, $T_{ETA}$ experienced a 67.7% difference (84.5–hour difference) at Edisto Island between Matthew and Dorian, while the peak $\eta_T$ belonged to the same categories during these two TCs. In addition, $T_{ETA}$ was dependent on $V_t$, $V_{max}$, and the distance to the TC eye.

### 5.5 Estimations of $R_2$ using different empirical formulas for $R_2$

In this study we used the empirical formula by Stockdon et al. (2006) to estimate $R_2$. However, there were different empirical formulas for the estimate of $R_2$. The wave breakers were categorized by Iribarren number ($\xi_0$ in Eq. 3). Stockdon et al. (2006) developed different forms of formulas of $R_2$ for the scenarios $0.3<\xi_0<4.0$ and $\xi_0<0.3$. The range $0.3<\xi_0<4.0$ represented the intermediate to more wave reflective scenarios. Senechal et al. (2011) proposed another formula based on the regression of

their observed data to improve the estimation under highly wave dissipative and saturated scenarios induced by the infragravity swash (Eq. 4). This included both the conditions with lower Iribarren numbers ($\xi_0<0.3$) and the extreme storm condition.

We compared the $R_2$ estimated by the formulas of Stockdon et al. (2006) and Senechal et al. (2011) at the same locations considered previously (Edisto Island, South Carolina; Sea Island, Georgia; and the barrier island south to Matanzas Inlet, Florida) during the three historical hurricanes (Fig. 11). The difference in the peak $R_2$ estimated by the two formulas reached

up to 1.35 m at the barrier island south of Matanzas Inlet estimated by Stockdon's formula, which was 76% higher than those estimated by Senechal's formula during Matthew and Dorian.

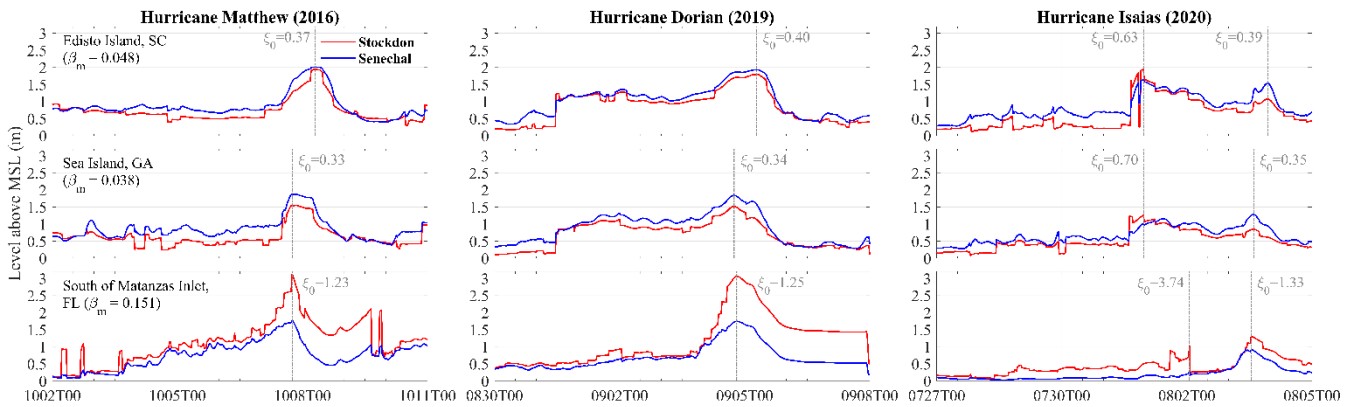





**Figure 11. The $R_2$ estimated by the formulas of Stockdon et al. (2006; red curves) and Senechal et al. (2011; blue curves) at three coastal sites during three hurricanes. The Iribarren numbers ($\xi_0$) corresponding to the peak $R_2$ (vertical grey dash lines) were listed.**

The $R_2$ estimated by Stockdon's formula showed a distinctive pattern during Isaias: another peak occurred before the storm approached the observed location. The difference in wave runup estimated by the two formulas reached 0.75 m, and the instantaneous Iribarren numbers were 3.74, 0.70, and 0.63 at the three coastal sites during Isaias. This was related to Isaias's unique path and faster $V_t$. The $R_2$ by Senechal's formula did not show this pattern, since Senechal et al. (2011) did not include the effect of $L_0$. The results showed that Stockdon's approach returned a larger $R_2$ compared to Senechal et al. (2011)

especially under $\xi_0>0.6$ (with a difference up to 1.34 m). In contrary, Senechal et al. (2011) gave larger values of $R_2$ under $\xi_0<0.5$ compared to the results by Stockdon et al. (2006) but with a smaller difference (i.e., < 0.50 m).

We computed the differences between the time series of $R_2$ derived from the formulas of Stockdon et al. (2006) and Senechal et al. (2011) (i.e., $\delta_R = R_{2-Stockdon} - R_{2-Senechal}$) along the SAB. Next, we calculated the $C_{CORR}$ between this difference and four parameters: $\xi_0$, $H_0$, $L_0$, $T_P$, and $\beta_m$. The results showed the difference between the two formulas had higher correlation

coefficients with $\xi_0$ ($C_{CORR}$=0.47), $\beta_m$ ($C_{CORR}$=0.64), and $L_0$ ($C_{CORR}$=0.42). $\delta_R$ increased as the mean beach slope, the offshore wavelength, and the Iribarren number increased, which resulted in more wave reflective conditions. Stockdon's approach predicted that $R_2$ increased as wavelength increased under certain conditions (faster $V_t$ and TC path allowing the swell to propagate toward the coasts). Further research and field measurements are needed in this direction given the relevance of accurately estimated *TWL*s and their durations.

**6 Conclusions**

We used the coupled ROMS–SWAN modeling system to simulate $\eta_S$ and the wave fields (wave energy spectrum and bulk wave parameters) within the SAB during Matthew 2016, Dorian 2019, and Isaias 2020. COAWST model results showed good agreement with the observed astronomic tides, surge, and wave heights. Following Serafin and Ruggiero (2014), we used the measured $\eta_0$ and waves to estimate the *TWL*s from observations. We used the linear wave theory to calculate the deep–water

wave parameters and estimate the 2% exceedance wave runup using Stockdon (2006)'s empirical formula. We followed the same procedure with the results from COAWST and compared the *TWL*s estimated with both methods.

We used our model results to compare the $\eta_T$ components at three coastal sites. The instantaneous $V_{max}$ and the distance to the hurricane eye were the key factors determining the peak $\eta_T$ within the near–TC wave field, whereas the maximum continual duration of $\eta_T$ and *TWL*s over given thresholds were primarily determined by $V_t$ and the distance to the hurricane eye. The

contributions of wave runup (i.e., wave setup (16% to 38%) + wave swash (41% to 57%)) to the peak $\eta_T$ was usually higher than $\eta_S$ (17% to 40%) at the three selected coastal sites during the three historical TCs. The variability of $\eta_S$ (up to 75%) at the peak $\eta_T$ under different TC properties was larger than that of the wave runup (wave setup + wave swash; ≤59%). These wave–dependent parameters were not only functions of the TC characteristics but strongly depended on the local coastal morphology (e.g., beach slope). The $\eta_T$ time series revealed that with specific TC characteristics (e.g., path, heading direction, and $V_t$) the





maximum $\eta_T$ may occur before the storm's peak (i.e., outside the near–TC wave field). This was observed in the case of Hurricane Isaias as the hurricane travelled with a fast instantaneous $V_t$ (maximum of 9.81 m s$^{-1}$ and average of 6.74 m s$^{-1}$, which was 1.1 to 2.3 times of the global average in all categories) two to three days before approaching the location.

Two empirical formulas of wave runup estimation were compared. Stockdon's formula predicted the extreme pre-storm swells associated with TC's faster translation speeds, whereas this peak was not observed when using Senechal's empirical formula.

Since runup observations during these storms were unavailable, it was not possible to determine which empirical formula was giving the best predictions. More observations of the wave runup during TCs are needed for the verification and calibration of wave runup parameterizations. With the present analysis of historical storms, it was difficult to determine the individual effect of each TC characteristic on *TWL*s. Further numerical experiments and analysis employing synthetic and idealized TCs are needed to quantify the individual impacts of $V_t$, $V_{max}$, distance to the coast, and beach slope on *TWL*s and, thus, the coastal

morphological impacts.

**CRediT authorship contribution statement**

Chu-En Hsu: Conceptualization, Formal analysis, Investigation, Writing – original draft, Validation, Visualization. Katherine Serafin: Methodology, Supervision, Writing – review and editing. Xiao Yu: Supervision, Writing – review and editing. Christie Hegermiller: Methodology, Software, Writing – review and editing. John C. Warner: Methodology, Software, Writing – review

and editing. Maitane Olabarrieta: Conceptualization, Funding acquisition, Methodology, Supervision, Writing – review and editing, Visualization.

**Competing interests**

The authors declare that they have no known competing financial interests or personal relationships that could have appeared to influence the work reported in this paper.

**Acknowledgements**

M. O. and C.-E. H. acknowledge the support from NSF thru the NSF Career-Award 1554892 and the USACE Engineering with Nature project; M. O. and J. C. W. acknowledge support from the National Oceanographic Partnership Program (Project Grant N00014-21-1-2203). This support is gratefully acknowledged.

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





## Appendices

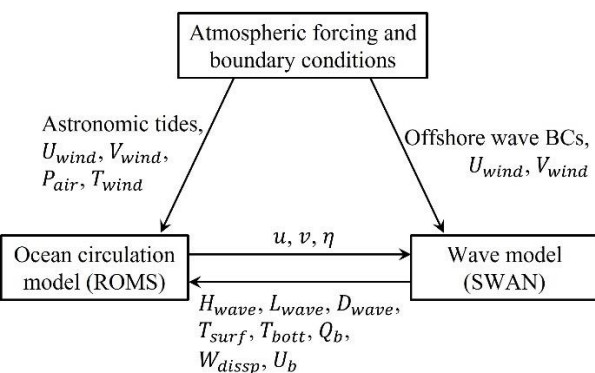

**Figure A1. Computational flowchart of ocean circulation-wave coupling using COAWST modeling system.**




**Figure A2.** A) Best–tracks of the three TCs (from the National Hurricane Center HURDAT2 best–track database) and computational domain and bathymetry (magenta: Matthew; green: Dorian; blue: Isaias; black: the boundaries of computational grids; hypsometric map: water depth). Panels B), C), and D) illustrated the best–track of the TCs (dots position), with the colormap of circles representing the maximum sustained wind during Hurricanes Matthew, Dorian, and Isaias, respectively. The time interval between adjacent dots was 6 hours.





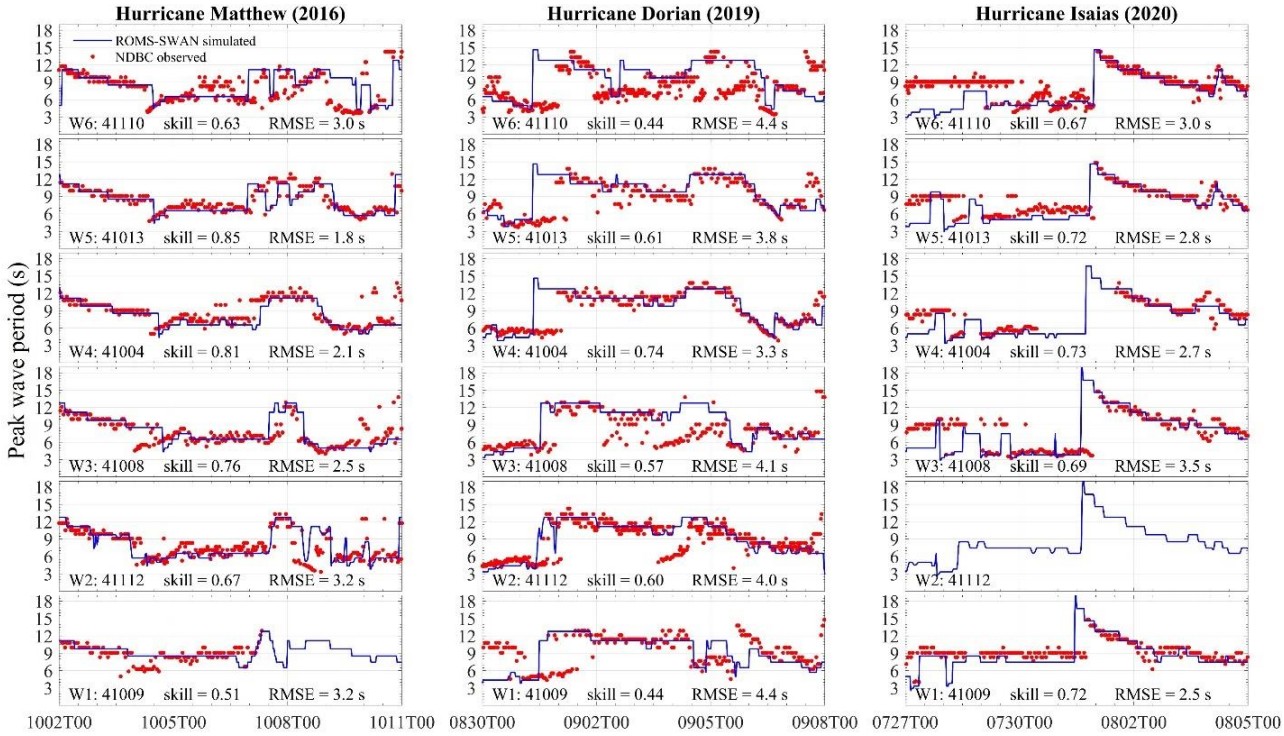

**Figure A3. Peak wave period ($T_P$) time series at six NDBS wave buoys during the three historical hurricanes. *Skill* was calculated with the formula of Willmott (1981), and *RMSE* denoted the root-mean-square error between model results (blue curves) and observations (red points).**




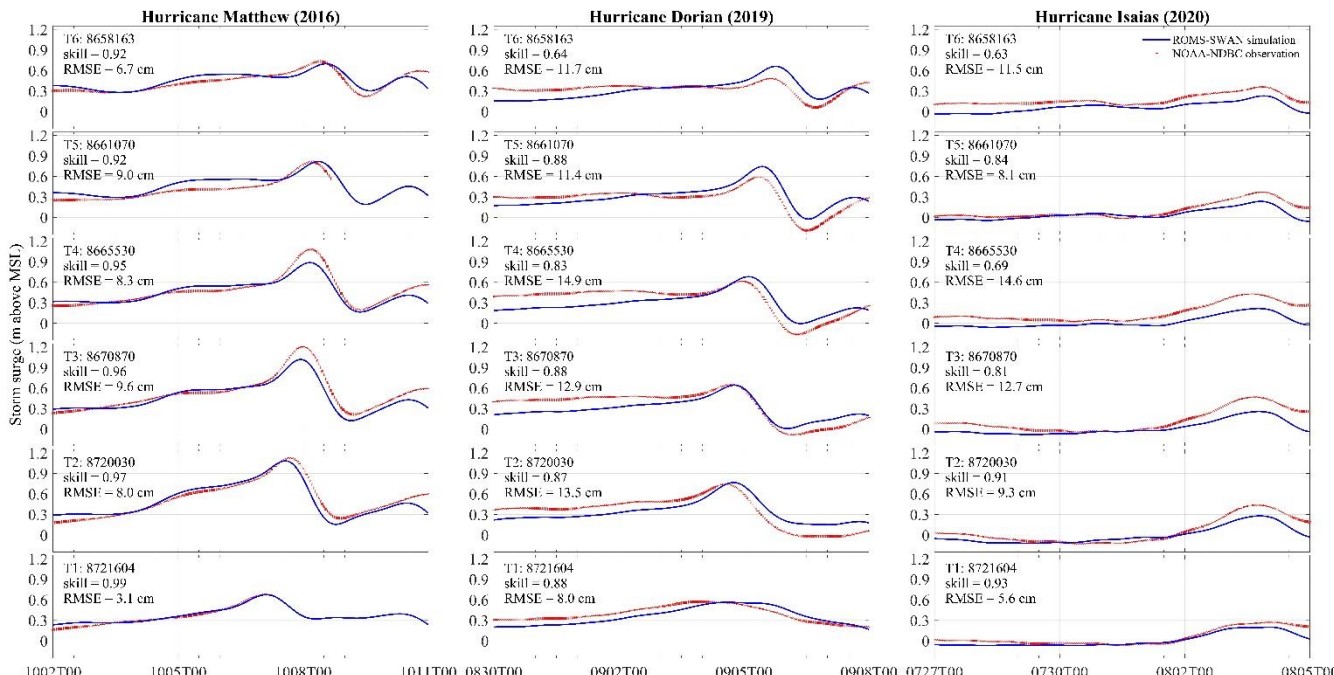

**Figure A4. Storm surge ($\eta_S$) time series at six NOAA tide gauges during the three historical hurricanes.** *Skill* **was calculated with the formula of Willmott (1981), and** *RMSE* **denoted the root-mean-square error between model results (blue curves) and observations (red points).**

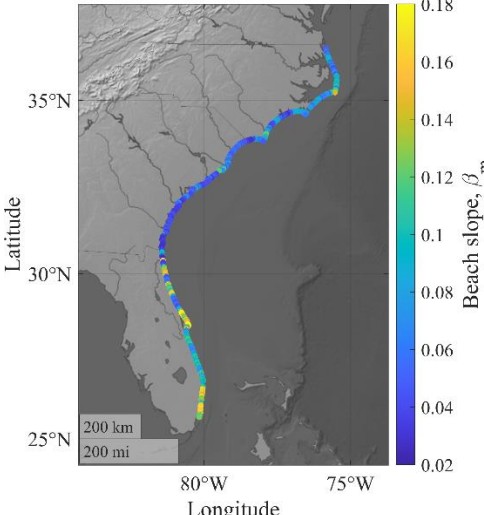

**Figure A5. Pre-Matthew mean beach slopes along the SAB measured by USGS.**

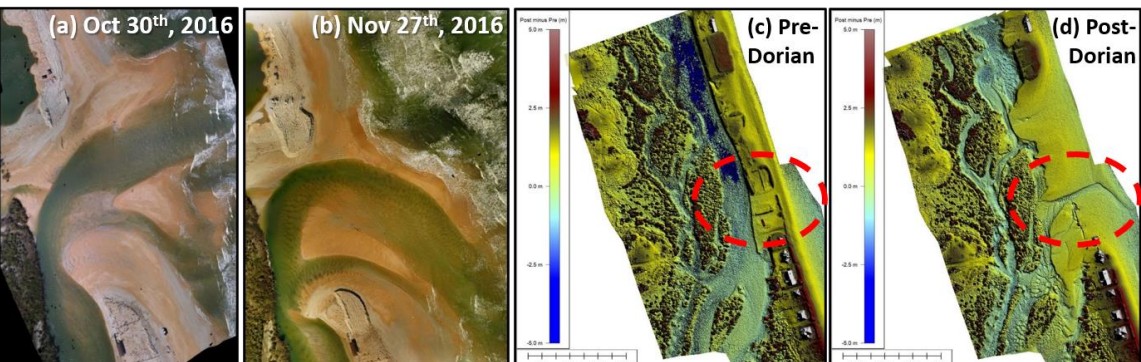

**Figure A6. Aerial images taken at the barrier island south of Matanzas Inlet, Florida after Hurricane Matthew, before Hurricane Dorian, and after Hurricane Dorian. The images were created by the authors.**