# Peer review of "Total water levels along the South Atlantic Bight during three along-shelf propagating tropical cyclones: relative contributions of storm surge and wave runup"

_Natural Hazards and Earth System Sciences, 2023_

## Author Comment (AC2)

**Response to Referee 1 Comments**

Dear reviewer,

We appreciate the time and effort you and the other reviewers expended in providing valuable feedback on our work. We considered and addressed all of the reviewer's suggestions. Changes to the manuscript have been marked.

Regarding the suggested citation for tidal distortion around Cape Canaveral, we thank Juan Felipe Paniagua-Arroyave for sharing their work and have included it in our introduction. As we receive and summarize all of the comments, we will upload the corrected version to the reviewers and editors.

**Point 1**: I suggest to avoid references in the abstract section because it should be stand alone. In this section, you must briefly introduce your personal contributions instead of the work of others.
**Response 1**: We have modified this part and removed the references from the abstract according to the reviewer's comment.

**Point 2:** As the manuscript stands today, some of the methods used in this application are in the introduction (e.g. the empirical formulations for R2). It would be helpful to have a methods section easily accessible to the reader within the manuscript. Moreover, some findings are introduced in the discussion section (see e.g. Figs. 10 and 11). I suggest merging the results and discussion sections into one section. Alternatively, if you want to keep this setup, please use this section only to discuss your findings.
**Response 2:** We have, accordingly, reorganized the structure of the manuscript. Everything related to methods is now in the methods section, and we have merged Results and Discussion into one section.

**Point 3**: Please, introduce the acronyms in the main text. For example, the translation speed of storm (Vt) is introduced in the table caption but not in the main text. Please, uniform the style throughout the manuscript.
**Response 3**: We have, accordingly, checked this issue throughout the entire document and have modified it.

**Point 4**: Avoid using symbols as +, -, or = inside brackets introduced in the main text, unless you are using an inline equation such as in line 418.
**Response 4**: We have, accordingly, modified this part.

**Point 5**: Sometimes you use "peak $\eta Ts$" and other times "peak $\eta T$". Be consistent throughout the text.
**Response 5**: We have, accordingly, modified this part.

**Point 6:** Figures 2, 3, 7, 11, A4, and A3. They lost quality during the exportation step. Lines, dots, axes labels, internal and external box grids are blurred. Please, improve the resolution.

**Response 6**: We have modified and re-plotted all the figures to make sure their resolution is good in the manuscript.

**Point 7**: Lines 39-42. It would be helpful to summarize the classification definition of the four regimes. There is the reference for more details. Maybe something like "[…] regimes: inundation (TWLs ≥ Dcrest), swash (TWLs < Dbase), overwash (…), and collision (…). Among these, coastal dunes experience […]" is more appropriate.
**Response 7**: We have modified this part and summarized the introduction for the four regimes more appropriately.

**Point 8**: Line 53. Avoid "=" inside the brackets. For example, you can say "(i.e. the sum of astronomic tides, mean sea level, and storm surge). You can also indicate the name of the variable you introduced. I suppose is the water level observed.
**Response 8**: We have checked and modified this throughout the entire document.

**Point 9**: Line 54. If data are available at the NOAA repository, please add reference and link.
**Response 9**: We have added the website link for reference.

**Point 10**: Line 66. Use "Eqs. 1 and 2" instead of "Eq. 1 and Eq. 2".
**Response 10**: We have checked for similar experssions in the document and modified them.

**Point 11**: Lines 70-71. Avoid the use of "=" inside brackets. Use instead "is".
**Response 11**: We have checked for similar experssions in the document and modified them.

**Point 12**: Line 72. Change Eq. 2 with Eq. 1.
**Response 12**: Changed.

**Point 13**: Lines 92-97. I suggest moving the statement "Thus, we employed […] for analysis and comparison" after "Despite the importance of […] had not been thoroughly examined". Also, please add "on the contrary" or "conversely" before "Senechal et al. (2011)".
**Response 13**: Changed.

**Point 14**: Lines 98-103. Please, reformulate the statements. For example, "Under projected climatic conditions, TC characteristics are projected to change at global scale (add reference). Thereby the proportion of high-intensity […] to increase. In this context of change, the main goal of this study is to ascertain how TC […] to TWLs. With that aim […] similar tracks".
**Response 14**: We have, accordingly, reformulated the statements.

**Point 15**: Line 108. The acronym Vt is introduced in Table 1 but not in the main text.
**Response 15**: We have added the definition of Vt in the main test.

**Point 16**: Line 145. Explain how you evaluate the suitability of the chosen exchange intervals.
**Response 16**: We have pointed out the relevant citations that used the same data exchange interval.

**Point 17**: Lines 152-153. Please, move the link after RAP and add the reference.

**Response 17**: We have added the reference for RAP dataset.

**Point 18**: Lines 155 and 162. Please, add references for the GFS and HYCOM database.

**Response 18**: We have added the references for GFS and HYCOM databases.

**Point 19**: Lines 165-168. Add reference for the Flather boundary. Furthermore, panel A is indicated with a capital letter in the main text and in the caption of the figure while in the figure a lowercase letter is used. Please, uniform the style.

**Response 19**: We have added the reference of Flather boundary conditions.

**Point 20**: Line 186. Please, add the repository link and references for NOAA tide gauges and NDBC buoys.

**Response 20**: We have added the website links for NOAA and NDBC data.

**Point 21**: Line 219. Tp is not yet introduced in the text.

**Response 21**: We have added the definition of Tp in the main text.

**Point 22**: Caption of Figure 4. I suggest using "the red, cyan, and black points indicated […]".

**Response 22**: We have modified the way this caption is written.

**Point 23**: Table 2. Improve the table headers. I suppose that Dbase refers to the second column.

**Response 23**: We have modified and re-arranged the table headers.

**Point 24**: Lines 276-277. As reported in Figure 5, the difference of 22.6 hours is for TETA. Please, change each of the contents in the brackets.

**Response 24**: We have changed each of the contents in the brackets.

**Point 25**: Figures 7 and 11. The x-axis label is missing.

**Response 25**: We have added the x-axis label.

**Point 26**: Figure 8. I suggest to introduce this figure in the supplementary material.

**Response 26**: We have moved this figure to the appendix and modified the corresponding description in the main text.

**Point 27**: Figure 10. Use A1, A2), B1, B2) and C1, C2) in the figure caption.

**Response 27**: We have modified the figure caption.

**Point 28**: Line 455. Change "in contrary" with "on the contrary".

**Response 28**: We have checked similar issues throughout the document and modified them.

Along with the above remarks, all grammar and spelling mistakes identified by the reviewers have been fixed. Regarding our application, we eagerly await your response and look forward to answering any additional inquiries and comments you might have. We appreciate the reviewer's comments and thank you for the feedback.

Sincerely,

Chu-En Hsu, Ph.D.
Postdoctoral Research Associate
University of Florida

---

## Author Comment (AC3)

**Response to Referee 2 Comments**

Dear reviewer,

We appreciate the time and effort you and the other reviewers expended in providing valuable feedback on our work. We considered and addressed all the reviewer's suggestions. Changes to the manuscript have been marked.

Regarding the suggested citation for tidal distortion around Cape Canaveral, we thank Juan Felipe Paniagua-Arroyave for sharing their work and have included it in our introduction. As we receive and summarize all the comments, we will upload the corrected version to the reviewers and editors.

**Point 1**: The manuscript compares coastal sea level setup associated with three different tropical cyclones that were similar in their intensities and trajectories, yet resulting in different coastal hazard along the South Atlantic Bight. As for other reviewer, for me the manuscript is also too techical and therefore hard to follow. Also, I am supporting other comments of other reviewers, so will not repeat them here.

**Response 1**: We have modified the writing and the structure of the manuscript to improve this part according to the reviewer's comment.

**Point 2:** In addition, what seems as a weakness to me is the model evaluation, which is performed integrally over a prolonged (10 days) interval with just one measure coming out of it ('skill'). However, such a defined skill cannot say anything about model performance during extreme conditions (i.e., during the close passage of a hurricane) as is largely determined by a backgound conditions to which models are normally performing better. From figures, I see (and not easy to see in figures) substantial underestimations and overestimations for some stations for peak sea level values. Therefore, I would like to see model verification during extreme $\eta 0$ and TWL values (using max values, or using q-q plots, tails of pdfs, or similar) with model performance validated during extreme conditions. Such an evaluation might also use estimates of 'maximum consecutive duration' from both model and observations, to see if the model can reproduce persistence in coastal flooding as well.

**Response 2:** We have, accordingly, utilized the correlation coefficients to quantify the model performance on the peak water level components and the maximum consecutive duration (see Fig. 2).

Along with the above remarks, all grammar and spelling mistakes identified by the reviewers have been fixed. Regarding our application, we eagerly await your response and look forward to answering any additional inquiries and comments you might have. We appreciate the reviewer's comments and thank you for the feedback.

Sincerely,

Chu-En Hsu, Ph.D.
Postdoctoral Research Associate
University of Florida

---

## Author Response (AR2)

**Response to Editors and Referees**

Dear editors and reviewers,

We appreciate the time and effort you expended in providing valuable feedback and handling all the processes for our submission. We have considered and addressed all the suggestions provided by the editors, reviewers, and scientific community. As we summarized all the comments, we have uploaded the final revised version to the NHESS editorial office.

Regarding our submission, we eagerly await your response and look forward to answering any additional inquiries and comments you might have. We appreciate the editor and reviewers' efforts and thank you once again.

Sincerely,

Chu-En Hsu, Ph.D.
Postdoctoral Research Associate
University of Florida